# HoaKV: High-Performance KV Store Based on the Hot-Awareness in Mixed Workloads

**Jingyu Liu [1], Xiaoqin Fan [1], Youxi Wu [1], Yong Zheng [2] and Lu Liu [3,*]**

[1] School of Artificial Intelligence, Hebei University of Technology, Tianjin 300400, China; liujy@hebut.edu.cn (J.L.); 202122802058@stu.hebut.edu.cn (X.F.); wuc@scse.hebut.edu.cn (Y.W.)
[2] State Grid Energy Saving Service Co., Ltd., Beijing 100052, China; zhengyong@sgecs.sgcc.com.cn
[3] School of Computer Science, Beijing Institute of Technology, Beijing 100081, China
[*] Correspondence: liulu@bit.edu.cn

**Abstract:** Key–value (KV) stores based on the LSM-tree have become the mainstream of contemporary store engines, but there are problems with high write and read amplification. Moreover, the real-world workload has a high data skew, and the existing KV store lacks hot-awareness, leading to its unreliable and poor performance on the highly skewed real-world workload. In this paper, we propose HoaKV, which unifies the key design ideas of hot issues, KV separation, and hybrid indexing technology in a system. Specifically, HoaKV uses the heat differentiation in KV pairs to manage the hot data and the cold data and conducts real-time dynamic adjustment data classification management. It also uses partial KV separation technology to manage differential KV pairs for large and small KV pairs in the cold data. In addition, HoaKV uses hybrid indexing technology to index the hot data and the cold data, respectively, to improve the performance of reading, writing, and scanning at the same time. In the mixed read and write workloads experments show that HoaKV performs significantly better than several state-of-the-art KV store technologies such as LevelDB, RocksDB, PebblesDB, and WiscKey.

**Keywords:** key–value store; LSM-tree; hash indexing; hot-awareness; KV separation

## 1. Introduction

Persistent KV stores are an essential part of modern store infrastructure [1,2]. KV stores are used in a wide variety of applications due to its excellent horizontal scalability and access speed and support for unstructured data stores, such as web search [3–6], e-commerce, social networking, data deduplication [7], and graph stores [8]. KV stores, organizes, and manages data in the form of KV pairs, usually providing a set of simple interfaces for data operation: write, read, and scan. With the development of Internet applications, the scale of user access and data are growing rapidly. Compared with relational databases, KV stores can better support mass user access.

The Log-Structured Merge-tree (LSM-tree) [9] is the main structure of persistent KV stores, such as the classic Google LevelDB [10]; Facebook RocksDB [11], which is a multi-threaded improvement based on LevelDB; Amazon DynamoDB [12]; the Apache-distributed database Cassandra [13]; the large-scale KV store HBase [3]; and BigTable [4]. The LSM-tree is a persistent index structure optimized for write-intensive workloads. The core idea is to improve write efficiency by sacrificing partial read performance and converting random write requests into sequential writes. The LSM-tree has the advantages of efficient write performance, efficient range query performance, and scalability. Compression operation is the key technology to ensure the read speed of the LSM-tree, but a large number of compression operations will reduce the system performance and lead to write amplification, which has always been the main limitation of the LSM-tree. Therefore, the previous research directions of KV store optimization based on the LSM-tree mainly include write amplification, compression operation optimization [14], adaptation to new hardware problems [15], special workload [16], secondary index or memory optimization [17,18],

etc. Reducing the write amplification [19] is often accompanied by a decline in query performance or the use of large memory. Thus, the full performance potential of KV stores are still constrained by the inherent multi-level-based LSM-tree design.

Among KV store workloads in the real world, delete-intensive and update-intensive workloads dominate many store scenarios, including server log cleaning [20] and online transaction processing [21]. Therefore, hot issues in mixed workloads (that is, a small number of projects frequently visited in highly skewed workloads) [22] are a common problem in real scenarios and have been extensively studied in the literature [23,24]. Many store systems, such as LevelDB [10] and RocksDB [11], use KV stores in memory to manage hot projects. UniKV [25] takes the latest incoming data as the hot data and uses the hash index in memory to index the hot data, to achieve efficient access performance. HotRing [26] is optimized for massively concurrent access to a small portion of items and dynamically adapts to the shift of hotspots by pointing bucket heads to frequently accessed items. However, how to accurately judge the hot and cold of the currently accessed file block has always been a research difficulty. In recent years, with the development of machine learning technology, various classification algorithms have been increasingly applied to the field of system structure design. Therefore, the use of machine learning algorithms, to predict the hot and cold of the file block and is applied to the cache optimization mechanism, is a problem worth studying. In addition, the access frequency of data changes dynamically, so it is meaningful to take timely response measures to the access of data.

To this end, we design a novel KV store. Based on the differentiated key-value management scheme, mixed index method, and a variety of well-designed technologies, HoaKV achieves high read, write, and scan performance for large KV stores with mixed workloads. Our main contributions are summarized as follows:

- We propose HoaKV, which coordinates the differential management of the hot data and the cold data to effectively adapt to mixed workloads. Specifically, HoaKV divides KV pairs according to the frequency of read and write access (i.e., heat), preferentially allocates system resources for the hot data to achieve fast access, and further carries out special management for the cold data according to its size.
- We propose a dynamic adjustment technology for the hot and cold data to achieve high scalability in large KV stores. Specifically, we timely adjust the classification of data and change its store management method according to the heat of real-time data changes.
- We propose a hybrid index method, namely the three-level hash index method in memory designed for the hot data and the three-level direct index technology on disk designed for the cold data, to improve I/O performance and reduce memory overhead.
- We propose a fine-grained partial KV separation and distinguish between small and large KV pairs in the cold data management to reduce the I/O overhead caused by frequent value movement caused by the compression operation of large KV pairs in the LSM-tree. In order to improve the performance of reading, writing, and scanning, we also propose a dynamic value grouping method to effectively manage the large KV pairs.
- We implemented a HoaKV prototype on LevelDB [10] and evaluated its performance using micro-benchmark and YCSB [24]. For micro-benchmark testing, HoaKV achieved efficient loading throughput, compared to LevelDB [10], RocksDB [11], WiscKey [19], and PebblesDB [27]. It also achieved significant throughput improvements in updates and reads.

## 2. Related Work

HoaKV is based on the previous work of building and optimizing KV stores. This section briefly introduces the previous work and the work close to HoaKV.

LSM-tree. Many persistent KV stores are built on the LSM-tree to solve scanning and scalability problems. In addition to building KV stores on new hardware such as non-

volatile memory or describing the real-world KV store workload, some research focuses on optimizing the write performance of the LSM-tree KV store, including optimizing the structure of the LSM-tree [27,28], KV separation [19,29,30], and reducing the compression overhead [31,32]. The main problem of write performance is the write amplification caused by the merge operation. For this reason, many researchers focus on how to optimize the merge operation. This can be roughly divided into two directions. One is the separation of key and value. WiscKey [19] uses the KV separation strategy to directly write the value into the value log and write the key and its corresponding value address into the LSM-tree. Helen H.W. Chan et al. put forward HashKV [29] based on WiscKey. Its core idea is to use the hash to group KV pairs, store the KV pairs in the corresponding segment group, and use the segment group as the unit when performing GC, thus reducing the GC overhead. However, its write performance is not ideal. Another research direction is reducing the write amplification by relaxing the requirement of data ordering in the same layer. DiffKV [28] utilizes a new structure, vTree, for value management with partial ordering. PebblesDB [31] proposes a fragmented LSM-tree, which relaxed the complete sorting of KV pairs by dividing each level into multiple non-overlapping segments and allowing KV pairs in each segment to not be sorted. UniKV [25] unifies hash indexing and the LSM-tree in a single system and leverages data locality with a layered design and dynamic range partitioning.

Hot-awareness. HashKV [29] proposes a distinction strategy between hot keys and cold keys. HashKV stores the hot keys in the segment of vLog, and separates the cold key, stores in the disk then. HotRing [26] proposes a novel hotspot-aware KVS, named HotRing, which is optimized for massively concurrent access to a small portion of items. Based on the cost–benefit model, uCleaner [33] proposes a method to separate the hot and cold data to reduce the I/O traffic caused by the phenomenon of valid data movement during GC.

Hybrid indexing. UniKV [25] aims to simultaneously achieve high performance in read, write, and scan, while supporting scalability, and it is also deployable in commodity store devices (e.g., SSDs). Data Calculator [34] and Continuums [35] focus on unifying the major different data structureto achieve self-designed KV stores. HiKV [36] and NoveLSM [37] designed a new index structure for nonvolatile memory. KVS_Top [38] uses a combination of hash and b-tree technologies to support the high-speed search of a large number of keys (40 million). DPPDL [39] adopts a dynamic partial-parallel strategy, which dynamically allocates the storage space with an appropriate degree of partial-parallelism according to performance requirements.

HoaKV also adopts the mixed index method. Different from the above hybrid index technology, HoaKV aims to achieve high-performance read, write, and scan, and supports scalability at the same time. HoaKV combines log structure and KV stores based on hash and sorting and uses a compact hash table to reduce the memory usage of each key. That is to say, HoaKV divides the data into the hot data and the cold data. Different methods are used for indexing the hot data and the cold data. In order to achieve fast read/write performance of the hot data, we use the hash index in memory. At the same time, for the index of the cold data, we use a common index that does not consume memory resources.

## 3. HoaKV Design

We propose HoaKV, which divides KV pairs into the hot data and the cold data, and further divides them into the large KV pairs and the small KV pairs according to the size of the cold data to achieve differential management of KV pairs. It supports efficient read and write through the hash index and the normal index. The data classification is adjusted through the dynamic change of the key value to the heat, to realize the dynamic scalable and high-performance KV store.

### 3.1. Architectural Overview

HoaKV consists of two parts as shown in Figure 1. The first part is called the HotStore, which stores the hottest part of KV pairs, and that is the data with the highest recent

read/write access frequency. The second part is called the ColdStore, which stores KV pairs differentiated by value size. Our insight is to calculate the heat of data using the read and write frequencies of KV pairs. The hot data are the largest part of the read–write ratio and accounts for only a small fraction of all KV pairs, so we keep them in the HotStore (fully sorting by heat) and index them directly with in-memory hash indexing for fast reads and writes. Meanwhile, we keep the remaining large amount of cold KV pairs in the ColdStore for efficient scans and scalability. HoaKV realizes the idea via the following techniques:

- Hot-awareness splitting. HoaKV stores the hot data in the HotStore. When KV pairs in a data block are written from in-memory, we calculate the heat of each KV pair and compare it with the minimum heat of the HotStore, then, if it is large, we store it in the HotStore.
- Hot KV indexing. To improve read performance, HoaKV stores the keys and values of the hot data separately. Also, HoaKV designs lightweight three-level hash hot indexing to balance memory usage and hash collisions. The hash indexing tables indexes keyTag, heat, and vTableID.
- Partial KV separation. To efficiently manage KV pairs in the ColdStore, HoaKV presents a partial KV separation scheme. The cold data are divided into small KV pairs and large KV pairs. Furthermore, a differentiated and fine-grained key-value management mechanism is implemented in the ColdStore to avoid frequent value movement in the merge process.
- Dynamic value grouping. To achieve high read and write performance in large KV stores, HoaKV proposes a value grouping scheme that dynamically splits KV pairs into multiple groups that are independently managed according to the key ranges, to expand a KV store in a scale-out manner.
- Cold KV indexing. In order to quickly find the location of the values of KV pairs and update the heat of the cold data in real-time, HoaKV uses the cold indexing table to record keys, heat, and group ID, where the value is located.

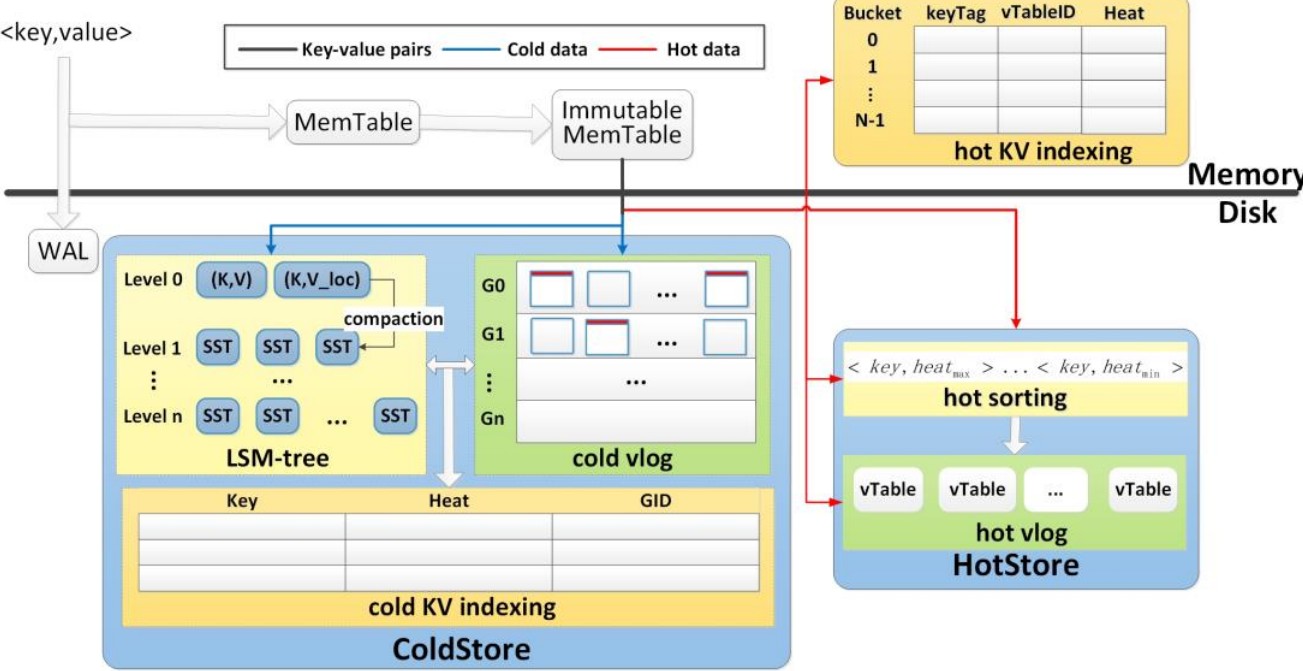

**Figure 1.** System overview.

### 3.2. Hot-Awareness Splitting

HoaKV divides all KV pairs into the hot data and the cold data. The hot data, which are the most frequently read–write accessed part of the data, accounts for a small portion of all key value pairs. The remaining data are the cold data, which accounts for the majority of the total data, and the cold data has less read and write access in a short time. HoaKV stores the hot data in the HotStore, and KV pairs are sorted by heat, that is, the data with the highest heat achieves the fastest read–write access efficiency. To reduce sorting overhead for the hot data, HoaKV stores the keys and values of the hot data separately.

We define the frequency of read–write access to KV pairs per unit time as heat, whose probability density function is as follows:

$$heat = (PW_i + PR_i)/T \tag{1}$$

where $PW_i$ and $PR_i$ represent the write frequency and read frequency of the *i*th KV pair, respectively, in the time *T*, and *T* is time. Data blocks are passed in from memory and the heat of each KV pair in the data block is calculated. HoaKV limits the size of the heat. When a KV pair has a heat greater than or equal to the predetermined threshold *HeatLimit*, the KV pair is the hot data, and HoaKV stores the KV pair in the HotStore. The KV pairs in the HotStore are sorted by heat, and the heat of KV pairs changes dynamically. The HotStore needs to sort the KV pairs frequently. To reduce sorting overhead, the HotStore stores the key and value of the hot data separately. The HotStore stores the key and the corresponding heat in the heat sorting and stores the value separately in the hot vlog.

Figure 2 depicts the hot-awareness splitting design. The data block are passed in from in-memory, and HoaKV updates and calculates the heat of each KV pair. If the heat of a KV pair is greater than or equal to *HeatLimit*, the KV pair is the hot data and are stored in the HotStore. For frequent read–write access to the hot data, the HotStore stores the keys and heat of the hot data in the hot sorting and sorts by heat. To reduce sorting overhead, the HotStore stores the values of the hot data in the hot vlog. The hot vlog is composed of multiple vTables, and the value of the hot data is stored in the vTable. The value of *HeatLimit* in the HotStore is dynamic. *HeatLimit* represents the minimum heat of the hot data. Due to the size of the HotStore being fixed, when the heat of the newly inserted KV pair is greater than *HeatLimit*, the KV pair corresponding to *HeatLimit* is redefined as the cold data, which is extracted from the HotStore and transferred into the ColdStore as the cold data. Then, the minimum heat in the latest sorting result is taken as *HeatLimit*, so the value of *HeatLimit* is dynamic.

The cold and hot data adjustment. The read and write access of KV pairs will increase the heat. The hot and cold data are not fixed, so they need to be adjusted dynamically. As shown in Algorithm 1, when a KV pair is inserted into the disk, we use a hash function to calculate the keyTag based on the key. Then, we search for the keyTag in the hot indexing table first. If it exists, it indicates that the KV pair is the hot data. We update its corresponding heat and perform a new hot sorting. The motivation proposed in this article is suitable for highly tilted workloads, so it is necessary for the efficient processing of the hot data. According to real-time reading and writing accesses of the data, we will timely update the location of the hot data in the disk through the heat sorting, so that the hot data that we need is accessed at the fastest speed and improves the performance of the storage engine. Furthermore, we require a higher time complexity for the sorting of the hot data, and the space complexity is not high. According to Algorithm 1, if the keyTag is in the hot indexing table, the system will update the heat of the hot data. As it is only updated and then sorted, the heat order sequence is basically orderly and decreasing, and the heat update of the hot data has been increasing. Therefore, based on the above characteristics, we have chosen the best sorting algorithm suitable for this situation for the hot sorting. Specifically, after updating the heat of the hot data, we start to compare and move forward from the position of the key to the previous node until the heat is less than the previous node. Therefore, in the worst case, the time complexity of this sorting algorithm is O

(N), and the space complexity is O (N). If corresponding to a read operation, the value corresponding to the key is returned from the hot vlog based on the vTableID in the hot indexing table. If it is a write operation, due to the real-time requirement for processing the hot data, we find the corresponding value from the hot vlog based on the vTableID and recycle the invalid value directly, where we then store the written value in the location of the old value. If the keyTag is in the cold indexing table and represents the KV pair as the cold data, we update its heat in the cold indexing table and compare it with *HeatLimit*. If it is greater than *HeatLimit*, we update it to the hot data. Specifically, we find the key with the lowest heat from the hot sorting, calculate the keyTag, find the vTableID from the hot indexing table, and then return the value from the hot vlog based on the vTableID. (this is the process of taking the heat minimum KV pair). Then, we update it to the cold data, compare its value to the threshold *Value_Size* (if greater than the threshold, take the key value separation technique, otherwise store the KV pair directly in LSM-tree). Then, we insert the new hot data into the hot sorting to further redo the hot sorting. Specifically, we calculate the keyTag of the new hot data, store the heat in the hot indexing table, store the value of the hot data in the vTable of the hot vlog, and then return the vTableID which is stored in the indexing entry of the hot indexing table. For the read operation, the value of the hot data is returned directly when the new hot data are inserted into the HotStore. For the write operation, we insert the latest value directly into the vTable of the hot vlog. If the keyTag is not in the cold indexing table, we insert it directly into the ColdStore. For read operation, we return the null values directly. For write operation, we write directly when the KV pair is inserted into the ColdStore. This enables dynamic adjustment of the cold and hot data.

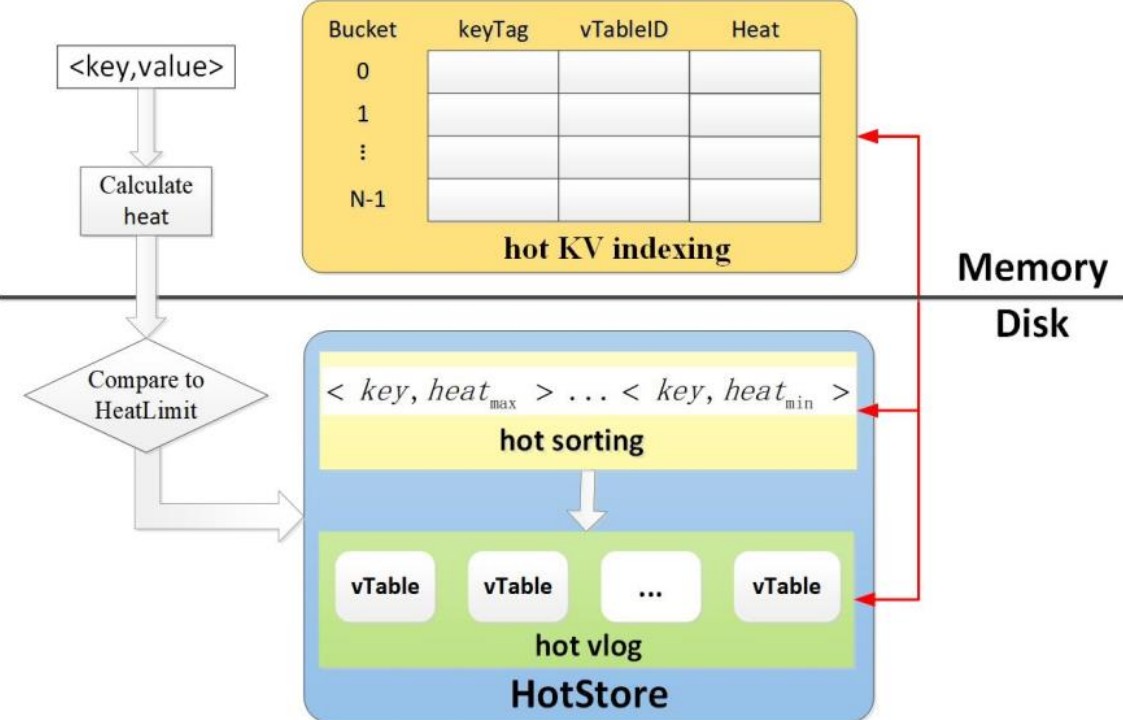

**Figure 2.** Hot-awareness splitting.

| Algorithm 1: Flow chart of dynamic adjustment of the cold and hot data |
|---|
| **Input:** KV pairs <key, value> |
| 1:    Calculate the keyTag |
| 2:    **if** the keyTag in the hot indexing table **then** |
| 3:            update heat in the hot sorting and the hot indexing table |
| 4:            adjust heat sorting |
| 5:            read the vTableID according to the keyTag |
| 6:            **if** operation == read **then** |
| 7:                    read value according to the vTableID in the hot vlog |
| 8:                    return value |
| 9:            **else if** operation == write **then** |
| 10:                find the value from the hot vlog based on the vTableID |
| 11:                recycle the invalid value |
| 12:                store the written value in the location of the old value |
| 13:            **end if** |
| 14: **else if** the keyTag in the cold indexing table **then** |
| 15:            update heat |
| 16:            **if** heat > *HeatLimit* **then** |
| 17:                take the heat minimum KV pair |
| 18:                calculate the keyTag |
| 19:                update it to the cold data |
| 20:                **if** value > Value_Size **then** |
| 21:                    take the key value separation technique |
| 22:                **else** |
| 23:                    store the KV pair directly in the LSM-tree |
| 24:                **end if** |
| 25:                insert the new hot data into the HotStore |
| 26:                redo the hot sorting |
| 28:                **if** operation == read **then** |
| 29:                        return the value of the new hot data |
| 30:                **else if** operation == write **then** |
| 31:                        insert the latest value directly into the vTable |
| 32:                **end if** |
| 33:            **end if** |
| 34: **else** |
| 35:            insert it directly into the ColdStore |
| 36:            **if** operation == read **then** |
| 37:                        return null |
| 38:            **else if** operation == write **then** |
| 39:                        write value |
| 40:            **end if** |
| 41: **end if** |

### 3.3. Hot KV Indexing

For data management in memory, HoaKV adopts a similar method to the traditional KV store based on the LSM-tree and ensures data durability using write-ahead logging (WAL). That is, the KV pairs are first appended to the log on the disk for crash recovery and then inserted into the MemTable, which is organized into a skiplist in memory. When the MemTable is full, it is converted into an Immutable MemTable. Then, according to the heat, a part of KV pairs, that is, the hot data, are refreshed to the HotStore on the disk via a background process.

Keys and values of KV pairs in the HotStore are stored separately; keys and the latest heat are stored in hot sorting via a heat-sorted manner; values are stored separately in the vTable of the hot vlog; and keys and values are indexed using a hash index in memory. To update and read the latest value in time, HoaKV also stores the heat in the hash index table. Its constituent level: <keyTag, vTableID, heat>. The keyTag stores the upper two bytes of the hash result calculated with the different hash functions. The vTableID is the location of

the hot data stored in the vTable of the hot vlog. The heat is the frequency of read–write access to KV pairs per unit time.

At the same time, in order to reduce the use of memory, HoaKV establishes a lightweight hash index, which uses a three-level hash. In addition, HoaKV uses the hash chain and cuckoo hashing method to solve the hash conflicts problem. As shown in Figure 3, the hash index contains $N$ buckets. Each bucket stores the index entries of KV pairs with cuckoo hashing, so it may append one or several overflowed index entries due to hash conflicts. When we create an index item for a KV pair, we search the bucket according to the hash results calculated using $N$ hash functions (from the general hash function library), i.e., $(h_1, h_2, \ldots, h_e, \ldots, h_E)$ $(key)\% N$, until we find an empty bucket. Note that we can use up to $N$ hash functions in this cuckoo hash scheme. If we cannot find an empty bucket among $N$ buckets, we will generate an overflow index entry and append it to the bucket located using $h_E$ $(key)\% N$.

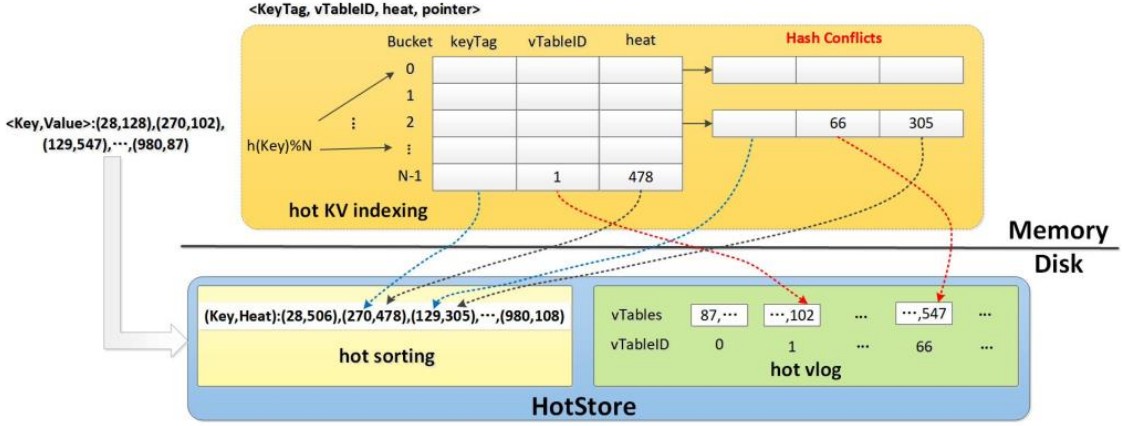

**Figure 3.** Hot indexing.

After finding a bucket, we record the keyTag and vTableID of the hot data in the selected index entry. Each index entry contains four attributes:<keyTag, vTableID, heat, pointer>. The keyTag stores the upper two bytes of the hash result calculated using the different hash functions, that is, $h_{n+1}$ $(key)$. It is used to quickly filter out index entries during key searching. The vTableID uses two bytes to store a vTableID. We can index 128 GB of vTables in the HotStore, each of which is 2 MB in size. The heat uses two bytes to store the latest value. The pointer uses two bytes to point to the next index entry in the same bucket.

The finding key and updating heat process works as follows: First, we use $h_{n+1}$ $(key)$ to calculate the keyTag. Then, we search for candidate buckets from $h_n$ $(key)\% N$ to $h_1$ $(key)\%$ $N$ until we find the hot data and update the heat. For each candidate bucket, the latest overflow entry is appended to the tail. Therefore, we compare the keyTag with the index entry belonging to the bucket from the tail of the overflow entry. Once we find a matching keyTag, we will use the vTableID to retrieve the metadata of the vTable andchange the heat value of the hot data. Note that due to the hash conflicts of $h_{n+1}$ $(key)$, the queried KV pair may not exist in this vTable, i.e., different keys have the same keyTag. Finally, if the KV pair is not found in the HotStore, it indicates that it is the cold data andwe need to further search in the ColdStore.

We now analyze the memory cost of the hash index. Each KV pair in the HotStore will consume an index entry, and each index entry will consume 8 bytes of memory. Therefore, for every 1 GB of hot data in the HotStore and the size of 1 KB KV pairs, it has about 1 million index entries. Considering that in our experiment, the bucket utilization is about 80%, it requires about 10 MB of memory. This memory usage is less than 1% of the data size in the HotStore. Note that for very small KV pairs, hash indexing may incur a large memory overhead. However, since all the data stored in the HotStore are the hot data,

that is, frequent read and write operations will occur in a short period, the large memory overhead caused by very small KV pairs is acceptable.

Our hash index scheme is a tradeoff in design. On the one hand, when we allocate buckets for KV pairs, there may be hash conflicts, i.e., different keys have the same hash value *h(key)* and are allocated to the same bucket. Therefore, we need to store the key information in the index entry so that the key can be distinguished during the lookup. On the other hand, storing a complete key wastes memory. In order to balance memory usage and read performance, HoaKV uses three hash values and reserves only a 2-Byte hash as a keyTag. This greatly reduces the probability of hash conflicts, which is also demonstrated in our experiment. Even if hash conflicts occur, we can still resolve them by comparing the keys stored on the disk.

### 3.4. Partial KV Separation

Recall that HoaKV stores a small number of hot data KV pairs in the HotStore and indexes with an in-memory hash index, which incurs additional memory overhead. The data that are not frequently read or written recently are defined as the cold data, which accounts for the majority of KV pair sequences. HoaKV stores the cold data in the ColdStore, that is, the data whose key value to heat is less than *HeatLimit*. As the size of KV pairs in the cold data is not uniform, if the large KV pairs data are directly stored in the LSM-tree, as in the traditional LSM-tree based KV store, it may cause great I/O overhead. As a result, the existing KV pairs in the LSM-tree need to be read and written back after merging. Therefore, how to reduce the merging cost of the cold data is a challenging, key problem for HoaKV. To improve the range query performance, HoaKV proposes a partial KV separation strategy, that is, the cold data are further divided according to its KV pair size; the key and value address of the large KV pair is stored in the LSM-tree; the value is stored separately in the cold vlog; and the key and value of the small KV pair are retained in the LSM-tree.

After the KV pairs sequence is split by the heat, the remaining KV pairs are the cold data, and we further classify the cold data. Depending on the size of the value, HoaKV categorizes the cold data as the small KV pairs and the large KV pairs. Differentiated fine-grained key-value management mechanisms are implemented for the different types of KV pairs. As shown in Algorithm 2, according to the threshold value which we set as *Value_Size*, HoaKV classifies the cold data KV pairs. Specifically, all KV pairs whose values are smaller than *Value_Size* are classified as the small KV pairs. KV pairs whose value is larger than *Value_Size* are classified as the large KV pairs. HoaKV uses different store and garbage collection mechanisms for different KV pairs. At the same time, HoaKV uses the heat index table on the disk to index the heat and the key and value for each KV pair.

---

**Algorithm 2:** Partial KV separation

---

**Input:** Cold KV pairs <key, value>
  1:   **if** value > Value_Size **then**
  2:      store value in the cold vlog
  3:      return value location
  4:      store key and value location in the LSM-tree
  5:   **else**
  6:      store key and value in the LSM-tree
  7:   **end if**

---

For the small KV pairs, HoaKV always stores the keys and values together in the SST file of the LSM-tree without KV separation. For the small KV pairs, KV separation will not bring obvious benefits, but will exacerbate issues such as read–write amplification and GC costs. The core of the key value separation technology is to store the key and the address of the value in the LSM-tree and store the value alone in the value log. For the garbage collection of the key value, which uses the key value separation technology, we need to find the corresponding address from the LSM-tree, find the value from the value log according

to the address, and then perform the garbage collection. This process increases the garbage collection cost of the storage system. Therefore, we need to reduce the cost of garbage collection as much as possible, and the garbage collection of the small key value is in the compaction process of LSM-tree. At the same time, due to the value of the small key value being relatively small, it has a small impact on the scale of the LSM-tree. This strategy reduces the system's garbage collection cost to a certain extent and retains the advantages of LSM-tree technology, including excellent insertion and search performance, and at the same time, alleviates the problem of I/O amplification. Therefore, the key value separation technology for small key values can lead to reading and writing amplification and GC costs. For the large KV pairs, HoaKV always performs a KV separation mechanism. HoaKV stores the value of the large KV pairs in the cold vlog, and the value in the LSM-tree is the location information of the value in the cold vlog. Therefore, for the large KV pairs, merging operations between levels on the LSM-tree only need to rewrite keys and metadata and do not need to move values, greatly reducing the write magnification of the large KV pairs.

### 3.5. Dynamic Value Grouping

With the data size growth of HoaKV, if we simply add more levels to large-scale stores as most existing LSM-tree based KV stores, moving data from a lower level to a higher level will lead to frequent compaction operations during write process, and trigger multi-level access during read process. Therefore, HoaKV proposes a dynamic value grouping scheme to expand the store horizontally. The scheme stores the values of the large KV pairs in different groups and manages them independently according to different key ranges.

The dynamic value grouping scheme works as follows (shown in Figure 4): Initially, HoaKV writes the value in a group (i.e., G0). Once the size of the group exceeds the predetermined threshold GroupSize, HoaKV will divide the group into two groups according to the key range and manage them independently (for example, G0 is divided into G0 and G1). For the value grouping strategy, the main feature is that the keys corresponding to the values stored in two groups are not overlapping. Therefore, how to split a group is crucial.

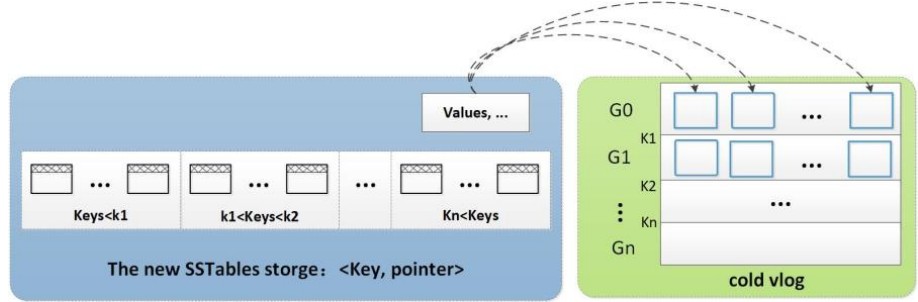

**Figure 4.** Dynamic value grouping.

To split the values in the cold vlog, HoaKV first locks them and stops the write request. Note that the unit of locking is a group, that is, HoaKV locks the entire group and stops all writes to the group during splitting. Then, it sorts all the keys corresponding to the values to avoid overlapping between groups. It first reads all the SSTable files related to the large KV pairs from the LSM-tree, sorts the keys, divides the sorted keys into two parts, and records the boundary key as *K* between the two parts. Note that the boundary *K* acts as a dividing point, that is, if the key of a large KV pair is less than *K*, its value is put into G0, and the remaining values are stored in G1. By dividing the points, HoaKV divides the values in the cold vlog into two groups whose keys do not overlap. Finally, HoaKV stores the value position with the corresponding key in the pointer, writes the key and pointer back to the corresponding SSTable file, and updates the value grouping information in the index table. HoaKV releases the lock and resumes processing the write request after splitting the value.

*3.6. Cold KV Indexing*

KV pairs in the ColdStore are managed differently by size. For large KV pairs, the KV separation strategy is implemented by storing the key and the address of the value in the LSM-tree and the true value in the cold vlog. For small KV pairs, the KV separation strategy is not implemented, and the key and value are stored directly in the LSM-tree. The data stored in the ColdStore are the cold data, but the heat of the data is changing. We need to adjust the store location and key-value management according to the heat of the data. We use an index structure to index the heat, key, and grouping number of the values.

To reduce disk usage, we build a lightweight index with three levels. Its constituent level: < heat, key, GID >. The heat is the frequency of read–write access to KV pairs per unit time. The key is the unique identification of the KV pair. The GID is the ID of the large KV pair in the cold data which is stored in the group. The hash index of the hot data uses the hash results calculated by the hash function to store the bucket. Unlike this, although the index structure also contains *N* buckets, it uses the direct indexing method. Considering that for large-scale store engines, if the direct indexing method is used purely, the search efficiency of the system will be reduced. Therefore, in order to speed up the search efficiency, we have improved the direct indexing method. The value of the cold data is stored in a dynamic grouping mode, so multiple values are stored in a group and the keys corresponding to these values are in the same range. Therefore, as shown in Figure 5, we store relevant information in the index structure according to the grouping sequence number (GID) of the cold data value, and HoaKV stores the heat information in the same bucket as the GID. Different values have the same GID, which will cause conflicts in the index structure. We use the link method to resolve conflicts. Therefore, one or more overflow index entries may be appended to each bucket due to the conflict of the GID. When we create an index item for a KV pair, we search the bucket according to the GID corresponding to the value. Note that in this scheme, if the bucket we find according to the GID is not empty, we will generate an overflow index entry and attach it to the bucket located by the GID.

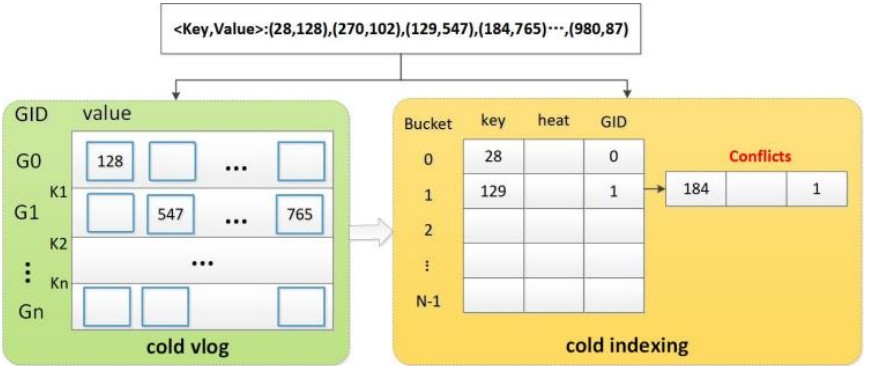

**Figure 5.** Cold indexing.

## 4. Evaluation

In this section, we evaluate the performance of HoaKV using real-workload-based benchmarks. In particular, we compare the throughput and scalability of HoaKV with several state-of-the-art KV stores: LevelDB, RocksDB, PebblesDB, and WiscKey. We also provide detailed evaluations to demonstrate the effectiveness of the major designs adopted by HoaKV.

*4.1. Setup*

We run all experiments on a machine with a 20 core Intel Xeon Silver 4210 2.20 GHz CPU which made by Intel Corporation from California, USA, 64 GB RAM, and a 4 TB SSD. The machine runs Ubuntu 20.04.6 LTS, with the 64-bit Linux 5.4 kernel and the ext4 file system.

For LevelDB, RocksDB, PebblesDB, and WiscKey, we use the same default parameters. Specifically, we set memtable_size as 64 MB (same as RocksDB by default), bloom_bits as 10 bits, and open_files as 1000. For block_cache_size, HoaKV sets it as 20 MB by default, while other KV stores set it as 170 MB to match the size of HoaKV's hash index for fair comparisons. The remaining memory is used as the page cache of the kernel. For the other parameters of different KV stores, we use their default values. For other parameters of HoaKV, by default, to balance write performance and memory costs, we set the group size to 40 GB. To limit the hash index in the HotStore, we set the size of the HotStore to 4 GB. For GC operations, HoaKV uses a single GC thread. In each test, if no other specification is made, we will use the default setting: 32 threads. We allow other KV stores to use all available capacity in our SSD RAID volume so that their major overheads come from read and write amplifications in the LSM-tree management. Finally, HoaKV uses YCSB [24] to generate various types of workloads. Generally, HoaKV sets the size of KV pairs to 1 KB and the key size to 24-Byte. HoaKV makes a request based on the Zipfian distribution, where the Zipfian constant defaults to 0.99 in YCSB.

### 4.2. Micro-Benchmarks

We evaluate the performance of the different KV stores, including the performance of load, read, update, and scan under the single-thread operations and the size of the KV stores. Specifically, we use YCSB to generate the workload and set the size of each KV pair to 1 KB, which consists of 8-Byte metadata (including key/value size fields and retention information), a 24-Byte key, and a 992-Byte value. We first randomly load 100 M KV pairs (approximately 100 GB). We then evaluate the performance of 10 M read operations, 100 M update operations, and 1 M scan operations that scan 50 GB of data. In addition, HoaKV sets some parameters, such as HotStore Size, Group Size, *HeatLimit*, and Value_size. During the evaluation of the micro-benchmarks, HoaKV takes the values of these parameters as: HotStore Size is 16 GB, Group Size is 20 GB, *HeatLimit* is 0.95:0.05, and Value_size is 32 KB.

Experiment 1 (the Performance of Load). We evaluate the load throughput for different KV stores and HoaKV. Figure 6a shows the load throughput of each KV store. Compared to other KV stores, it shows that HoaKV's load performance is 9.6 times that of LevelDB, 6.2 times that of RocksDB, 1.8 times that of PebblesDB, and 0.8 times that of WiscKey. It is important to note that HoaKV is implemented based on LevelDB, but its performance is higher than other specifically optimized KV stores except WiscKey. This is because WiscKey separates each KV pair, so the LSM-tree has the least amount of data and therefore has a higher load throughput than HoaKV. The load throughput of HoaKV is much greater than LevelDB because HoaKV uses partial KV separation. There are more KV pairs in the same layer, which also makes HoaKV have more I/O resources to service user requests, so HoaKV has a much higher load performance than LevelDB. Load performance is mainly affected by write amplification, so the comparison results of load performance are similar to those of write amplification.

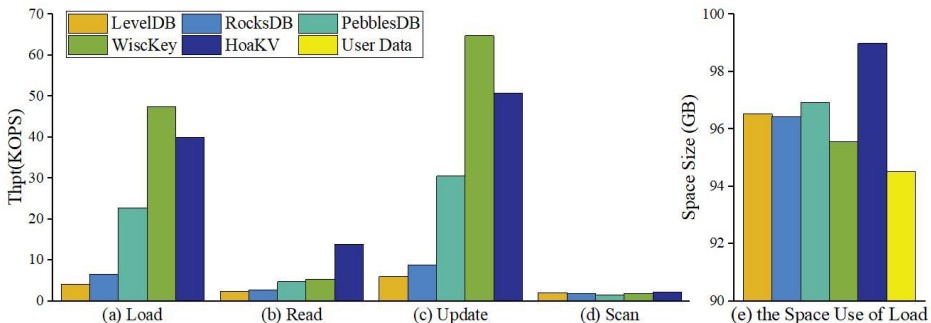

**Figure 6.** Micro-benchmark performance.

Experiment 2 (the Performance of Read). We then evaluate the performance of 10 M read operations on various KV stores. Figure 6b shows the throughput of each KV store

performing read operations. As can be seen, HoaKV has the best-read performance. The read performance of HoaKV is better than WiscKey, mainly because for HoaKV, the small KV pairs in the cold data do not perform KV separation, and there is no need to issue another I/O request during reading. The read throughput of HoaKV increases by nearly five times compared to LevelDB, which is also because HoaKV's differentiated key-value management strategy allows the LSM-tree to store more KV pairs per layer than LevelDB, with an average of fewer layers to search for a KV pair.

Experiment 3 (the Performance of Update). We evaluate the performance of 100 M update operations for different KV stores and HoaKV. As shown in Figure 6c, WiscKey has the highest update performance because it directly writes KV pairs to the value log without the need to update them to WAL files and memory tables. It also has the lowest update I/O volume and performs the best. HoaKV has a lower update performance than WiscKey, but its update throughput is 7.5 times higher than LevelDB. This is mainly because LevelDB's severe write amplification affects its update performance, while the write amplification problem of HoaKV is much better because it stores the value of the hot data in the hot vlog and stores the value of the large KV pairs in the cold data into the cold vlog.

Experiment 4 (the Performance of Scan). We also test the scan performance of various store systems. Figure 6d shows the scanning throughput of each store system. According to the results, LevelDB performs the best for scan operations as it stores all KV pairs in an orderly manner in the LSM-tree without performing KV separation. Compared with WiscKey which fully implements KV separation, HoaKV has a 12.5% improvement in scan performance. This is because most of the data in the LSM-tree is stored in the bottom two layers, while in the LSM-tree of HoaKV, small KV pairs do not perform KV separation, greatly improving scan performance.

Experiment 5 (the Usage of Space). Figure 6e shows the total KV store size for different KV stores after all load and update requests are issued. In addition, they have very similar KV store sizes, meaning that all systems consume similar store space during the loading phase. HoaKV incurs a slight additional store overhead, mainly used to store and record pointers to the value positions of the hot data and the large KV pairs in the cold data.

### 4.3. YCSB Evaluation

Experiment 6 (YCSB performance). Next, we evaluate the performance of various KV stores using the default workload of YCSB, which is an industry standard for evaluating KV stores. As shown in Table 1, YCSB provides four different core workloads (Workloads A-D), each representing a read–write mode in a real-world application scenario. Specifically, Workloads A and B are read–write mixed with 50% and 95% reads, respectively. Workload C is a read-only workload with 100% reads. Workload D also includes 95% reads, but reads queries for the latest values.

**Table 1.** YCSB Read/Update ratio.

| Workload | Workload A | Workload B | Workload C | Workload D |
|---|---|---|---|---|
| Read | 0.5 | 0.95 | 1 | 0.95 |
| Update | 0.5 | 0.05 | 0 | 0.05 |

We present the performance results of LevelDB, RocksDB, PePePebblesDB, WiscKey, and HoaKV under the default YCSB core workload. Figure 7 shows the total throughput of each KV store area under each YCSB workload. In both read–write-dominated workloads, HoaKV always performs better than other KV stores. In Workload A, compared to other KV stores, HoaKV is 4.7 times that of LevelDB, 1.2 times that of RocksDB, 3.2 times that of PebblesDB, and 2.2 times that of WiscKey, respectively. The performance of HoaKV and RocksDB is similar, mainly because under workloads with fewer updates, RocksDB no longer delays write operations to refresh the MemTable. It can better provide reads and updates through multi-threading optimization. Next, we consider Workload B, Workload C,

and Workload D, all of which are read-intensive. HoaKV is 4.4–11.4 times that of LevelDB, 1.2–3.0 times that of RocksDB, 2.3–7.4 times that of PebblesDB, and 1.0–2.5 times that of WiscKey, respectively.

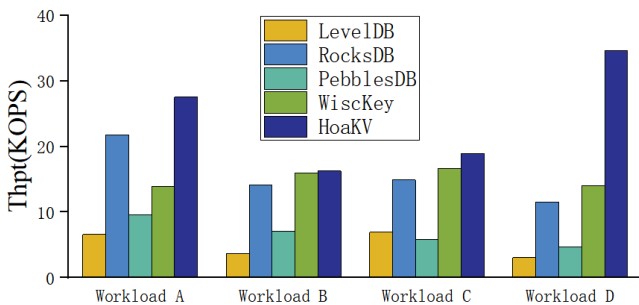

**Figure 7.** YCSB performance.

### 4.4. Performance Impact

Experiment 7 (Impact of the HotStore size). We investigate the effect of the HotStore size on HoaKV. We randomly load 100 M KV pairs and issue a 10 M read operation. Figure 8a shows the result of modifying the size of the HotStore from 1 GB to 16 GB when the fixed group size is 40 GB. As the size of the HotStore increases, the load throughput also increases, while the read performance remains almost unchanged. However, the memory cost of hash indexing for the HotStore will increase. Therefore, the size of the HotStore should be limited to balance performance and memory overhead.

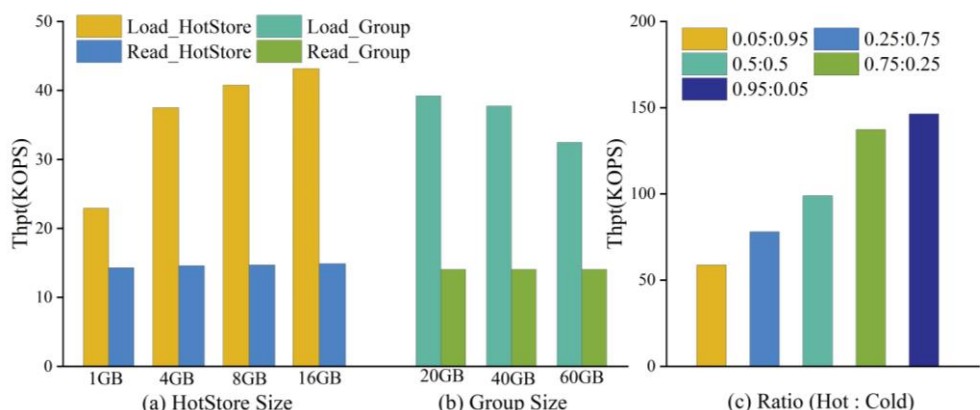

**Figure 8.** Performance Impact.

Experiment 8 (Impact of the Group size). We analyze the impact of the group size on HoaKV. We randomly load 100 M KV pairs again and issue a 10 M read. Figure 8b shows the result of changing the group size from 20 GB to 60 GB while fixing the store area size of the HotStore to 4 GB. The impact of group size on write performance is minimal, while it has almost no impact on read performance. The reason is that GC operates independently within each group. Therefore, the smaller the grouping, the more effective the GC operation. However, group size can affect memory costs, as HoaKV needs to allocate a MemTable for each group. Therefore, smaller groups may occupy more memory, so group size should be limited.

Experiment 9 (Impact of *HeatLimit*). We evaluate the impact of *HeatLimit* on HoaKV's update performance. The size of the heat threshold *HeatLimit* also represents the proportion of the hot data and the cold data in HoaKV, so we consider five different proportions of the hot data and the cold data, including 0.05:0.95, 0.25:0.75, 0.5:0.5, 0.75:0.25, and 0.95:0.05. Figure 8c shows the update throughput for the different ratios in the hot data and the cold data. Thus, as the proportion of the hot data becomes heavier, the update performance of HoaKV becomes higher. As the hot data indexes the keyTag, vTableID, and heat through

a hash index table in memory, HoaKV can quickly find the corresponding KV pairs and update them.

Experiment 10 (Impact of Value_Size). We investigate the effect of KV size ranging from 256 B to 32 KB and maintained other parameter settings. Figure 9 shows the throughput of randomly loading 100 GB KV pairs, reading 10 GB, and updating 100 GB KV pairs. To better illustrate the performance trend of data access, the throughput shown in this graph is in MB/s. As the size of KV pairs increase, both HoaKV and PebblesDB have higher throughput due to their efficient sequential I/Os. HoaKV always outperforms PebblesDB in terms of load, read, and update performance. When the KV pair becomes larger, the improvement of HoaKV reduces the throughput of loading the KV store and increases the throughput of reads and updates. The reason is that as the size of KV pairs increase, PebblesDB maintains more SSTables in the first level. This reduces compression overhead but can cause read operations to check these SSTables one by one, resulting in a decrease in read performance.

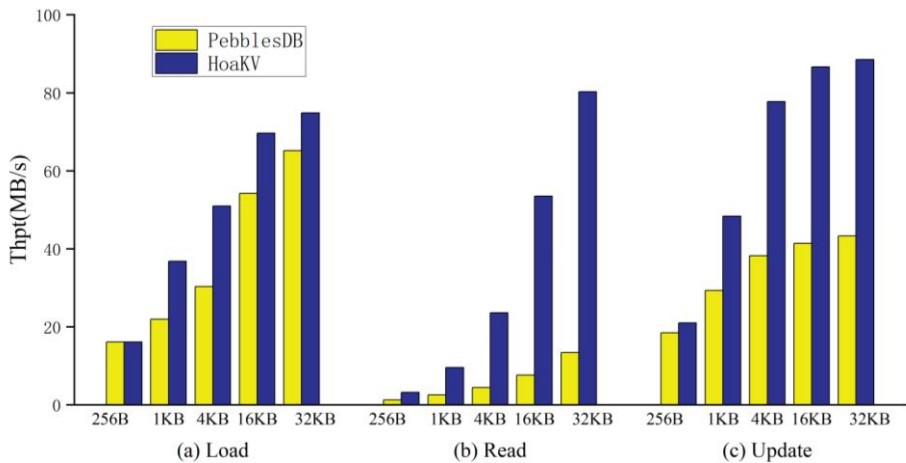

**Figure 9.** Impact of Value_Size.

## 5. Conclusions

In this paper, we propose HoaKV, which divides KV pairs into the hot data and the cold data, and further divides them into the large KV pairs and the small KV pairs according to the size of the cold data to achieve the differential management of KV pairs. It supports efficient read and write through the hash index and the normal index. The data classification is adjusted through the dynamic change of the key value to the heat, to realize the dynamic scalable and high-performance KV store. In HoaKV, the differentiated GC method is used for the two log files. Due to the unique characteristics of the hot data, the GC of the hot data requires timeliness. In order to reduce GC overhead, HoaKV proposes a delay method based on the number of invalid values in each packet of the cold vlog. The test experiment shows that HoaKV achieves efficient read, write, and scan performance and has low store cost. HoaKV achieves a balance of performance in all aspects.

Future research directions are as follows: the optimization of distributed KV storage systems. This article mainly focuses on optimizing KV storage systems on a single machine. For distributed KV storage systems, more issues need to be considered. In a distributed system, there may be load imbalance among nodes, which affects the overall performance of the distributed KV storage system. Therefore, we hope to conduct more in-depth research on data consistency and load balancing in distributed KV storage systems.

**Author Contributions:** Conceptualization, J.L., X.F. and Y.W.; methodology, J.L. and Y.Z.; software, J.L. and X.F.; validation, J.L., X.F. and L.L.; writing—original draft preparation, J.L.; writing—review and editing, J.L. and L.L.; supervision, Y.Z. and L.L. All authors have read and agreed to the published version of the manuscript.

**Funding:** This work was supported by the National Natural Science Foundation of China (no. U1936218 and 62072037). The funder is Chinese Government.

**Data Availability Statement:** The data presented in this study are available on request from the corresponding author. The data are not publicly available due to privacy.

**Conflicts of Interest:** The authors declare no conflict of interest.

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
