# Peer review of "HoaKV: High-Performance KV Store Based on the Hot-Awareness in Mixed Workloads"

_electronics, doi:10.3390/electronics12153227_

Round 1

Reviewer 1 Report

Advantages: 

This work aims to improve performance of KV store by considering KV pairs into hot and cold data. Additionally, with the cold data, this work divides KV pairs into large and small KV pairs in size. Those proposals support effectively reading and writing in KV store.

This menuscript is good at:

1.   Proposing quite good solutions for KV store

2.   Describing schemes in a coherent way 

3.   Considering experiments in almost cases (micro, macro benchmarks, various impacts of parameters, etc.)

Disadvantages: 

However, if following issues are better solved, it will be fully influential for this work.

1.   Are there any definition, explanation, comparison for T in equation (1)? How does this T impact heat values? Should the impact of T be shown in evaluation? 

2.   Equations should be expressed carefully in proper forms. For example, with equation (1), PWi, PRi are considered to calculate which “heat”? or “heat” of what? At line 259, the number of something is usually denoted by using upper cases (i.e., N, K, E, etc.,). Maybe, at 260, (h1, h2, …, he, …, hE) (key)%N looks better. 

3.   In algorithm 1, is this flow chart? 

 Moderate editing of English language required

Author Response

We are very grateful for your comments. We have provided a point-by-point response to the your comments and either upload it as a Word file.

Reviewer 2 Report

I have only formal comments:

Chapter 3 of HoaKV Design is empty and should be filled with text before the authors write subsection 3.1.
In Figure 6, part e is probably a typo - the sapce. Please make a correction.

Author Response

       We are very grateful for your comments. We have provided a point-by-point response to your comments and either upload it as a Word file.

Reviewer 3 Report

This manuscript introduces HoaKV, a novel key-value store with hot-data awareness to address the challenge of high data skew in practical workloads. HoaKV incorporates hot-data awareness as a distinguishing feature, enabling the differentiation of hot and cold data based on a predefined threshold named HeatLimit. Moreover, HoaKV implements real-time dynamic adjustment mechanisms and leverages partial KV separation technology, utilizing another threshold, Value_Size, to differentiate between large and small key-value (KV) pairs within the cold data. To demonstrate the efficacy of their design, the authors conducted a comprehensive evaluation, comparing HoaKV against prominent KV stores such as LevelDB, RocksDB, PebblesDB, and WiscKey.

Strengths:

The manuscript is well-organized and easy to comprehend. The authors provide a comprehensive overview of the overall design as well as detailed explanations of crucial techniques employed in HoaKV. These techniques include hot-awareness splitting, hot KV indexing, partial KV separation, dynamic value grouping, and cold KV indexing. By offering clear descriptions and explanations, the authors ensure that readers can grasp the concepts and techniques presented.

Moreover, the authors conducted an insightful evaluation that encompassed microbenchmarks and the widely adopted Yahoo! Cloud Serving Benchmark (YCSB). This evaluation enabled a thorough comparison between HoaKV and state-of-the-art KV stores, providing valuable insights into the strengths and weaknesses of different systems. Notably, the authors conducted a meticulous analysis of the impact of the two thresholds employed in their design, HeatLimit and Value_Size, and provided practical recommendations for parameter tuning. These comprehensive evaluations and analyses contribute to the substantial interest and relevance of this work for both academic researchers and industry practitioners.

Comments/Suggestions:

1. I recommend increasing the font size in Figure 1 for better readability. It would also be beneficial to include a legend in the figure, explicitly explaining the meanings of the colors used for the lines (blue and red, likely representing cold and hot data, respectively).

2. In Algorithm 1, the authors explained that if the keyTab is in the hot indexing table, the system will update the heat and adjust heat sorting. Could you the authors elaborate more on the specific sorting algorithm used here and the algorithm’s complexity?

3. In line 254, the authors referred to a three-level hash without explicitly enumerating its constituent levels. To enhance comprehension, I suggest presenting these three levels as bullet points or in a similar format.

4. In Lines 329-330, the authors claimed that KV separation would cause read-write amplification and GC costs. It would be beneficial to provide more detailed explanations and insights into the mechanisms behind these consequences.

5. Although the authors outlined the process of exchanging hot and cold keys in Algorithm 1, they did not delve deeper into the design aspects of this exchange. Given that such exchanges could occur frequently in practice, it is advisable to provide further elaboration on how these exchanges are resolved. Specifically, for instances where a large KV pair resides in the cold store and experiences increased access frequency, it would be valuable to discuss the mechanisms by which it would be transferred to the hot store. Additionally, it would be insightful to explore the potential impact of dynamic value grouping or compaction on these exchanges.

6. I would like to inquire about the consistency of the thresholds (HeatLimit and Value_Size) of HoaKV during the evaluation of the microbenchmarks. It would be helpful to clarify whether these thresholds remained constant or were varied throughout the evaluation process.

7.  It would be beneficial to include future work in the Conclusion, outlining potential directions for further research and development. This addition would provide readers with insights into the authors' vision for advancing the proposed solution.

N/A

Author Response

(The authors gave the same response as above.)

Round 2

Reviewer 3 Report

The authors have addressed all my comments. I suggest accepting the manuscript.

N/A